# Advances in Search Strategy Using the Set of Brand Considerations in the Web Ecosystem

**Sungeun Kwon** [1], **Jonghyuk Kim** [2] and **Zoonky Lee** [1],*

1   Graduate School of Information, Yonsei University, 50, Yonsei-ro, Seodaemun-gu, Seoul 03722, Korea; kjinju3@yonsei.ac.kr (S.K.); zoonk.lee@gmail.com (Z.L.)
2   Division of Computer Science and Engineering, Sunmoon University, 70, Sunmoon-ro 221 beon-gil, Tangjeong-myeon, Asan-si, Chungcheongnam-do 31460, Korea
*   Correspondence: zlee@yonsei.ac.kr; Tel.: +82-2-2123-4528

**Abstract:** This study explores changes in a set of brand considerations as a result of web search strategies. Survey and personal computer log data of car buyers were used to identify online information search behavior for brands and products. Through this study, we found that higher frequencies of brand searching are associated with how much consumer-initiated sites and third-party-initiated sites are used, while lower frequencies of brand searching are only related to how much brand-initiated websites are used. We also concluded that ambivalent messages on consumer-initiated sites lead to the postponement of a decision and a continued search for another brand. In addition, third party-initiated information sources lower search costs, which lead to longer consumer journeys and expand the set of brands considered and searched. The results of this study can help marketers understand the importance of their own media and aid in the development of a digital media strategy.

**Keywords:** web search strategy; brand considerations; online advertising; network analysis; Markov chains graph; brand-initiated website; consumer-initiated website; third party-initiated website

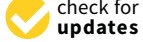



## 1. Introduction

In a digitized multimedia environment, consumers search for brand and product information online before purchasing [1–3]. Research by the Google Shopper Science team showed that 71% of potential car buyers searched for information online before visiting a dealership. As consumers are substituting Internet-based searches for traditional searches [1], numerous studies have examined consumers' search behavior on the Internet [1,4–9]. Furthermore, in today's online environment comprising numerous different types of information platforms, understanding information platforms and search behavior is important in planning a media strategy across a range of digital media [2,10].

There are two main groups of researchers in web search behavior studies. One is the consumer behavior (CB) community, which concentrates on psychological or cognitive factors related to search behavior from the perspective of shoppers [1,9,11–13]. The other is made up of information scientists who are concerned with user characteristics, search tasks, system capabilities, and behavioral patterns of web searches from the perspective of system users [10,14–16]. One of the common interests of both communities is the search strategy, that is, how searches are conducted. Many of the previous studies conducted by information retrieval (IR) researchers have usually focused on the search process or difference of search strategy depending on previous knowledge/experience, question factors, or user education instruments [10,15,16]. These researchers delivered a great deal of insight into how people search on the web, but most did not explain the relationship between search strategy and outcome, that is, how different results were incurred depending on the search strategy. Most IR research has lacked a sense of information system users as decision-makers (such as shoppers) and has focused on isolated parts of users' searches, while many

CB researchers are devoted to drawing a typology of search strategy [2]. Moreover, few of them observed search behavior from the beginning of the purchase process to its end. This is because their research was based on consumer surveys and interviews. Traditional approaches using retrospective consumer surveys or interviews have tried to measure perceived search behavior and the perceived number of brands searched. However, this type of data can produce only limited conclusions about consumers' behavioral patterns on the Internet [2]. For instance, consumers rarely remember their movements across digital information sources and the points at which their searches were longer than others.

Our goal was to explore consumers' web search strategies and the set of brand considerations as an outcome of the search strategy. First, we used personal computer (PC) clickstream data, which records each purchase-related website that a consumer visits before making a purchase, to create clusters of online search strategies based on the main websites of consumers' searches. Then, the number of brands searched online was analyzed for each search strategy type. The results will help marketers understand consumers' online search behavior and plan media strategies across different types of digital media. Furthermore, this study seeks to improve the theoretical model of consumer online search behavior and brand consideration.

## 2. Background

### 2.1. Search Behavior and Set of Brands Considerations

The Internet overcomes geographical distances and temporal limitations, so the cost of searching for information is not a problem nowadays. Today's consumers search with less cost pressure on the web. In internal search processes, experiences or previously acquired information is saved in the memory and recovered for a search. In an external search, information is acquired from various sources [17]. In general, consumers prefer to use internal information from memory when making decisions because it costs less. When the benefits of an internal search are insufficient, consumers turn to external sources [1]. In the external search process, the first step is to select the product or brand under consideration for purchase. The products/brands to be selected are known as the set of considerations. The second step is the final selection of the consideration set. Some products/brands among the set are unconsciously recalled from memory, while others are new products or brands encountered during the external search [17,18]. Whether a specific product or brand is included in a set of considerations is determined by comparing the cost of inclusion and the overall benefits. This comparison includes the cognitive costs expended in deciding whether to include a brand in a product family under consideration and the cost of evaluating its inclusion against other product families under consideration. According to shopping theory, as consumers proceed through the shopping process, their shopping purpose becomes clearer. The set of considerations at the beginning stage of shopping is diverse and easily changed according to external stimuli. However, at the later stages, as the purpose of shopping becomes clearer, these sets of considerations tend to be culled as consumers narrow their choices [19]. When browsing online at the beginning stage, the set of considerations has been changed by information filtration and information integration from diverse information sources [20]. This study focuses on the initial stage of shopping before the final selection of the set of considerations. We assume that stimuli from diverse online information sources can increase or decrease the number of products or brands under a set of considerations. In this study, we investigated which type(s) of online information sources stimulate changes in brands in the set of considerations.

On the other hand, the IR literature on search strategy suggests that if users have a high web experience level, they are more likely to adopt a "known address search domain strategy", while if their web experience level is low, they are more likely to adopt a "sequential player strategy", "search engine narrowing strategy", or "broad first strategy" [15]. The search strategy depends on question characteristics, such as clarity, complexity, and predictability. The researchers found that closed, clear, and predictable questions lead to a "direct address strategy", while complex, open, and unpredictable

questions lead to "subject directory or search engine strategy" [10]. Search assistance, such as peer advice or cognitive authority, also affects tactics in search moments or strategies in search sequences [16].

## 2.2. Types of Digital Information Sources

Various systems are used to classify the digital information sources. A study identified functional, social, community, and corporate digital information channels based on their content, direction, and interaction [21]. Functional channels are unilateral media from firms to consumers, and email is the most common example. Social channels are media operated by administrators and have the ability to limit users' access; these channels facilitate two-way communication between users and administrators or between users. Groups of users operate community channels in which participants interact and share content with others, such as blogs or YouTube. Corporate channels, such as brand websites, provide objective information such as corporate history, product specifications, product images, official prices, brand promotions, store locations, customer service, and answers to frequently asked questions. In contrast, social or community channels provide commentary or recommendations from buyers on their product experiences, product usage, actual prices, actual discounts, stores used, product defects, and answers to queries from potential buyers at the pre-purchase stage. Similarly, a study classified digital media as brand-owned, partner-owned, consumer-owned, and social/external touch points [22]. Another study classified two types of digital information sources to measure the effects of online advertisements on purchase conversions [23]. The first type is firm-initiated contact, such as brand homepages, in which content is created and messages are sent by firms. The second type is consumer-initiated contact, in which content is formed, and messages are accepted through consumer actions, such as search engines and price comparison sites.

In this study, digital information sources were classified as brand-, consumer-, and third-party-initiated media, depending on who operated the source. Brand-initiated media are operated by firms and deliver brand-generated messages, including brand homepages, brand microsites and blogs, and Facebook pages. Consumer-initiated media mainly communicate consumer-initiated messages, such as personal blogs, online communities, and general communities' message boards. Third party-initiated media generally support communication between firms and consumers and include a variety of sites, including search engines and retailer sites. In this study, used car dealers' sites, search engines, webzines, automotive-specific portal sites, and the automotive sections of portal sites were classified as third-party-initiated media. These digital sources were hypothesized to play different roles in the process of consumers' purchase decisions because they provide different information to consumers.

## 2.3. Role of Digital Information Sources and Set of Brand Considerations

Many models of consumers' search behavior have been proposed in the information-seeking behavioral literature. The models explain information-seeking behavior through psychological theories, such as gap bridging, stress/coping, risk/reward, cost–benefit, social learning, and uncertainty theories [15,24]. These theories provide excellent descriptions of the psychological dynamics of the routes consumers take to reach their purchase decisions. According to theories of uncertainty, consumers search for information to reduce their uncertainty about brands. There are two types of brand uncertainties: individual and relative. Relative brand uncertainty is about which brand is best, whereas individual brand uncertainty is what each brand offers [25–27]. Considering the theories and studies on the uncertainty of consumers' search and digital information sources together, we think that high individual brand uncertainty leads consumers to sources of brand-initiated messages such as brand homepages, brand blogs, or consumer testimonials. On the other hand, we assume that high relative brand uncertainty sends pre-purchase buyers to third-party information sources such as webzines, online retailers, and automotive portals, where they obtain information on product comparisons or product evaluations by reliable authorities.

Risk theories also shed light on consumers' search behaviors. Information is a way to reduce perceived risk [28]. Perceived risk refers to consumers' subjective expectations of loss. Bhukya and Singh examined four types of risks and concluded that functional risk and financial risk were more negatively associated with purchase intention than with physical or psychological risks [29]. Cost–benefit theories help to explain why consumers seek information from purchasers or information sources such as social media or expert webzines [11,20]. For example, a study found that consumers visit social media sites to obtain information about experiences from post-purchasers, to learn about new products, and to make their final purchase decisions [30]. Another study revealed that consumers prefer to acquire information through others' experiences rather than through their own research because experience is often cheaper than searching [31].

Information from actual purchasers transforms experienced goods into search goods, providing people with an "experience without ownership" [32]. Researchers also found that experts from webzines or forums influenced users' paths to purchase [28]. Experts often offer helpful information, so they are considered more reliable than other sources of information. From the viewpoint of cost–benefit theories, users' experiences on social media or recommendations from experts provide a speedy and efficient way to reduce brand uncertainty and even the risks of purchases. Given the dynamic interactions among the different online media formats discussed above, Table 1 summarizes how brand-initiated media usually provide detailed information about their own products and brands.

**Table 1.** Information sources and their roles.

| Information Sources | | Key Information | Roles |
|---|---|---|---|
| Brand-initiated media | Brand's homepage | Brand history, product specification/image, store location, customer service, etc. | Learn about products, reduce individual brand uncertainty |
| | Brand's microsites and blogs | More detailed information about product specification, official prices, brand promotion, etc. | Learn about products, reduce individual brand uncertainty |
| | Brand's Facebook page | Brand news, promotions, fans' comments, etc. | Learn about brand news and events |
| Consumer-initiated media | Personal blogs | Personal experience, usage scene of product, actual price, etc. | Learn about products efficiently |
| | Online car community | Sharing experience from actual buyers, Q&A between potential buyers and actual buyers, store information, product defects, etc. | Reduce individual brand uncertainty, reduce risk of purchase |
| | General car community's message board | Information on diverse brands' events and news. | Learn about brands and products |
| Third party-initiated media | Used car dealers' site | Product comparison, resale price, etc. | Reduce relative brand uncertainty and risks |
| | Search engine | Links to bridge information sources, etc. | Learn where to search |
| | Webzine | Expert reviews, recommendations, etc. | Learn about products from reliable parties |
| | Automotive-specific portal | Product comparison, product specifications and images, etc. | Reduce relative brand uncertainty |
| | Automotive section of a portal site | Product comparison, product specifications and images, etc. | Reduce relative brand uncertainty |

This information plays a role in conveying ideas about each product and, therefore, reduces individual brand uncertainty. Consumers who want information about product quality or to read about the experiences of actual users visit consumer-initiated sites.

Information from other users can help consumers to learn quickly and efficiently about products and brands while serving to reduce the perceived risks. Third party-initiated sources also play important roles in reducing relative brand uncertainty by providing information about alternative products and experts' product evaluations. We conducted a study comparing various search strategies related to CB. To the best of our knowledge, a comparison of web search effectiveness has not been attempted in past studies, and we aimed to determine the most strategic search method based on consumer clickstream data from a vast number of relevant sites. In addition, by studying which search method is most effective for brand search according to the situation of consumers, it will be possible to determine practical implications for companies or public institutions enacting related legislation. Based on the theoretical foundations of consumers' search behavior and previous studies about a variety of information platforms on the Internet, we proposed the following research questions (RQs): 1. What type of search strategy is identified based on an information platform on the web? 2. Is there a difference in the number of brands searched among the search strategy types?

## 3. Methods

### 3.1. Sampling

We bought PC clickstream data from a commercial panel service company with over 300,000 panelists. The clickstreams contain URLs, visit times, and routes of the websites that consumers visit prior to their purchases. The gap between time stamps was calculated to determine the amount of time the panelists spent on each site. This study is based on data on new car purchases. Automobiles are high-involvement purchases that motivate potential buyers to expend considerable effort in research [27]. For this reason, automobiles have often been the subject of research on consumers' information search behavior [12,20,27,31]. To collect purchase process data, panelists were first given a survey to identify which of them had bought cars between December 2018 and November 2019. The survey asked about the date of purchase and the make and model of the purchased vehicles. Over 200 panelists responded to the survey. The PC clickstreams of over 200 panelists were identified as including automobile-related sites. This was accomplished by compiling a list of 433 automotive-related websites. Using this list of target sites for analysis, we isolated the panelists' clickstreams related to automobiles. Data from survey respondents whose digital search behaviors were unidentified or who visited fewer than three sites were removed from the dataset, and the final analysis was based on 84 panelists. These panelists' clickstreams were then filtered to include only the three months preceding the dates on which they purchased their automobiles.

However, our unit of analysis was not simply a sample of 84 panelists, but the clickstreams of the online sites they visited. Therefore, we believe that the study methodology was sound, in that we used a rather complex and versatile method for analyzing smaller consumer clusters. In addition, we tried to increase statistical accuracy by repeating the modeling process. To achieve this, we divided the data into two sets: the training data set, which included the first part of the total data, and the validation data set, which included the remaining data. The clickstreams indicated that there were 117,091 visits to 433 target sites. We then calculated the duration of visits at the site, frequency, and sequences for each panelist. The final number of URLs included in the analysis was 11,628, which included 17 brand homepages, 8 brand microsites, 5 brand blogs, 4 brand Facebook sites, 23 personal blogs, 177 online car communities, 2 general community car billboards, 45 used car dealers' websites, three search engines, 23 webzines/automobile sections on Internet newspapers, 10 automotive-specific portals, and four automotive sections of a portal site. Every URL in the data was classified into one of the 11 digital media categories. These 11 categories were grouped again into the three dimensions of digital information sources in Table 1, which were used to cluster the car buyers who pursued the same online search strategies. To extract the number of brands, the clickstream data provided rich information such as the brands on websites in cases of brand-owned websites or brand-related online communities

and search terms on search engines, let alone types of websites. We extracted brand names from the URLs and classified them based on brand and the online car community. With these data, we were able to add new variables for the brands searched and the number of brands searched for every panelist. For the survey, we identified automobile buyers and asked them about makes and models that were candidates for purchase, the way to compile a set of considerations, the budgets they set before starting their research, the actual purchase price they paid, information about the car they ultimately purchased, and demographic information about themselves.

To answer RQ1, the time spent on the three dimensions of digital information sources was added as a new variable. Buyers normally used only 2–3 digital information sources among 11 categories, and there were differences in the total time spent on individual purchasers. Thus, the time spent on 11 categories of digital information sources was summed up into new variables representing the three dimensions of digital information sources and the time spent on brand-initiated sources, consumer-initiated sources, and third-party-initiated sources. These three new variables were normalized for statistical analyses. First, online search behaviors were segmented based on the time spent on the three dimensions of online sources. As in prior research on online browsing and media behavior, K-means clustering was applied to the group search strategies and time spent on the three types of informational sources [33,34]. As a result of this clustering, buyers' search strategies were classified into three groups, as shown in Table 2. Then, to understand buyers' movement between digital information sources, network analysis was applied to visualize the consumer flow, and the Markov chain approach was introduced to calculate the probabilities of visiting digital information sources during the path to purchase [35,36]. Finally, to answer RQ2, Markov chain analysis and the analysis of variance (ANOVA) were applied to examine differences in the number of brands searched among the three search strategy groups.

**Table 2.** Characteristics of informational source usage.

| Share of Time Spent for | Group 1 (n = 23) | Group 2 (n = 27) | Group 3 (n = 34) | Sig. |
|---|---|---|---|---|
| Brand-initiated sources visited | 69.4% | 13.5% | 6.1% | ** |
| Consumer-initiated sources visited | 9.2% | 61.4% | 9.6% | *** |
| Third party-initiated sources visited | 23.4% | 25.1% | 84.3% | ns |
| Sum | 100.0% | 100.0% | 100.0% | |
| Average of total time spent (min/person) | 185.8 | 1377.8 | 673.1 | ** |

(Significant Level: ** $p < 0.05$, *** $p < 0.01$)

*3.2. Descriptive Analysis*

RQ1 asks what characteristics of digital media usage are associated with each type of search strategy. Table 2 presents the characteristics of the three search strategy groups. The size of the three search strategy groups was 27.4 % (23 panelists), 32.1% (27 panelists), and 40.5% (34 panelists). From the point of total time spent viewing, Group 1 spent the least amount of time spent searching online, while Group 2 spent the most time (Group 1: 185.8 min/person > Group 2: 673.1 min/person > Group 3: 1377.8 min/person). As for the time spent on the dimensions of informational sources, Group 1 used brand-initiated informational sources the most, devoting 69.4% to their messages. The third party-initiated sources were the second most important informational source for Group 1 at 23.4%. Group 2 used consumer-initiated sources the most, with 61.4% of consumer-initiated information sources such as online car communities. As for Group 3, its members spent 84.3% on third-party websites, which was the most among the three search strategy groups. Considering the time spent on the main informational sources, Group 1 was termed the brand message-oriented group, Group 2 was termed the consumer message-oriented group, and the third was named the third party-oriented group.

Table 3 presents the descriptive statistics of the three search strategy groups that were analyzed using survey data from the same panelists with PC Clickstream. To identify "representative" buyers for each search strategy group, we compared the demographics and the characteristics of the search strategy groups. The brand message-oriented group tended to be older than the other groups at first glance, and the consumer message-oriented group appeared to be younger. However, a closer inspection of the results revealed that the probability of early tentative conclusions was less than 0.1, which means that this sample provided no evidence to support such a conclusion. Moreover, the typical users of this consumer message-oriented group appeared to be predominantly male (96.3%). However, according to the ANOVA test, the conclusion about gender characteristics showed a very weak significance level ($p < 0.1$). In terms of automobile characteristics, the consumer message-oriented group appeared to favor less expensive cars than the other search strategy groups, while the consumer message-oriented group favored smaller cars and the third-party message-oriented search strategy group favored larger cars. However, the interpretation of car price and car capacity also failed the significance test, according to the ANOVA test. Only the characteristics of car brand origin were tested as significantly different among groups ($p < 0.05$).

**Table 3.** Characteristics of demographics and automobiles by search strategy group.

| Cluster Size | Brand Message-Oriented Search Strategy (n = 23) | Consumer Message-Oriented Search Strategy (n = 27) | Third-Party Message-Oriented Search Strategy (n = 34) | Sig. |
|---|---|---|---|---|
| Demographics | | | | |
| % of male buyers | 73.9% | 96.3% | 76.5% | ns |
| Mean of age | 38.4 | 35.0 | 37.7 | ns |
| Automobiles | | | | |
| % of imported car brand buyer | 4.4% | 7.4% | 26.5% | ** |
| % of large-car buyers [1] | 21.7% | 3.7% | 20.6% | ns |
| Mean of car price | $27,314 | $23,686 | $29,953 | ns |

(Significant Level: ** $p < 0.05$) [1] Large car means automobiles with capacity of over 3000 cc.

## 4. Results

### 4.1. Characteristics of Online Path to Purchase by Search Strategy Group

To understand the information on the difference in behavior in the search strategy, we visualized the consumers' online purchase paths through Gephi, an open graph business platform for network analysis, as shown in Figure 1. We also used visit sequences and origin–destination pair set data to extract these customer purchase paths. Figure 1a displays a strong tendency to search for information on search engines, automotive sections of portal sites, and brand homepage sites. The group with a brand message-oriented search strategy was more likely to visit the homepage of the purchased brand than the homepage of other brands if their journey began with a search engine. Figure 1b reveals that the group with the consumer message-oriented search strategy moved differently from the search engine to the next destination. This group tended to move from search engines to consumer-initiated sites, such as online car communities, rather than to brand-initiated sites. Moreover, this group tended to search for information about considered brands more than purchased brands, as evidenced by the flow that they moved from the automotive sections of portal sites to the homepage of considered brands. The group with a consumer message-initiated search strategy tended to visit sites with consumers' voices or sites with considered brand information. Lastly, Figure 1c shows that the group with a third-party message-oriented search strategy tended to search for information on the automotive sections of portal sites and then move to used car dealers' sites. They also frequently moved from search engines to online communities of purchase brands and considered brands.

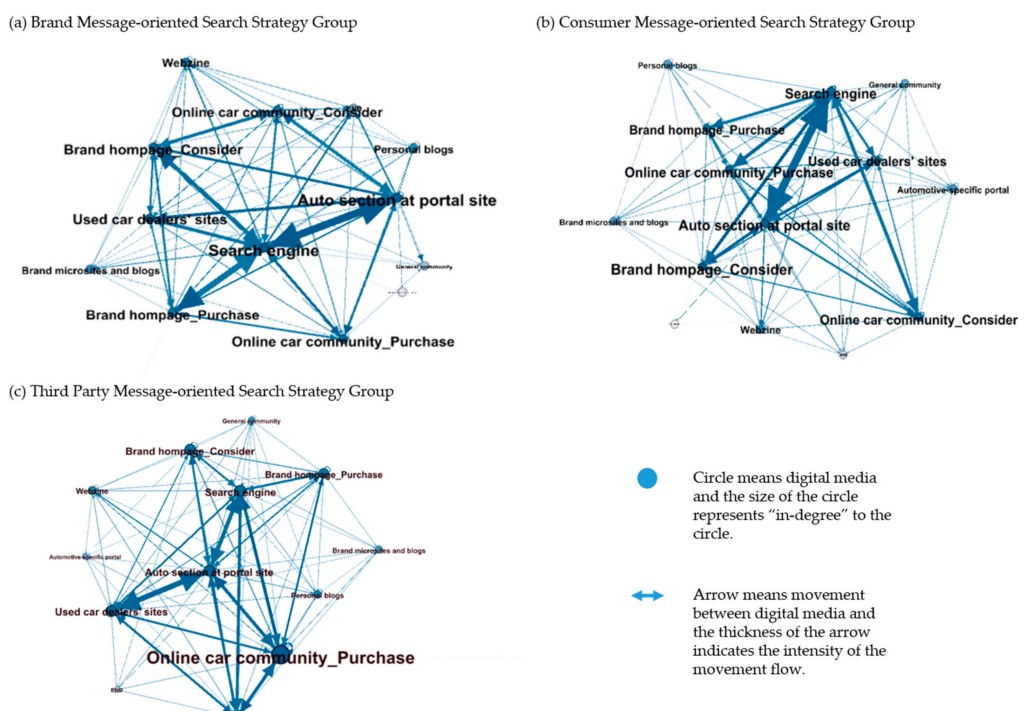

**Figure 1.** Visualization of consumers' online paths to purchase by network analysis.

### 4.2. Modeling Web Search Strategy with Markov Chains Graph

We discussed patterns in which panelists accessed websites. These patterns were determined by first computing the visiting sequence transition matrix for each consumer journey group. Then, the panelists' visiting sequences were modeled using Markov chain graph modeling, and the marginal probability of the transition was computed. The consumer decision journey is a process composed of a sequence of unpredictable events. Accessing certain types of online sources is correlated with the probability of making a purchase [35]. Markov chains allow for the identification of structural correlations in the customer journey and so have frequently been used in attribution modeling or process mining studies [36]. In addition, Markov chains provide the marginal probability that each digital information source would be visited during the journey. Markov chains were used to model the order in which panelists visiting websites and were defined as graph M = < S, W>, where S was the group of site categories and W was the transition matrix that contained the weighted probabilities of moving between website categories. The transition matrix W indicated the probability that the searcher would move between the digital information source categories.

$$W_{ij} = P(X_t = S_j \mid X_{t-1} = S_i), \quad 0 \leq W_{ij} \leq 1 \tag{1}$$

where $W_{ij}$ is the conditional probability of a panelist moving from the i-th digital information source category to the *j*-th digital information source category [36]. A stationary Markov chain based on Bayesian probability estimation was used in this study to obtain convergence by calculating the transition matrix, which was the key to understanding consumer journey patterns. To calculate the transition matrix, a dataset of origin–destination pairs for each group was compiled. Origin–destination pairs were obtained by determining the transitions between states in the visiting sequence.

Figure 2 shows an example of a sequence of website visit sequences. The consumer first visited a "used car dealer's site", then visited the "homepage of the brand of car that they ultimately purchased", and then visited "the homepage of a brand of car that they did not purchase". This sequence yields two pairs: used car dealer's site/purchased brand homepage and purchased brand homepage/considered brand homepage. The transition

matrix and Markov chains graph were developed based on these origin–destination pairs. The following section derives the journey maps for each consumer group and presents the convergence value for each type of consumer journey.

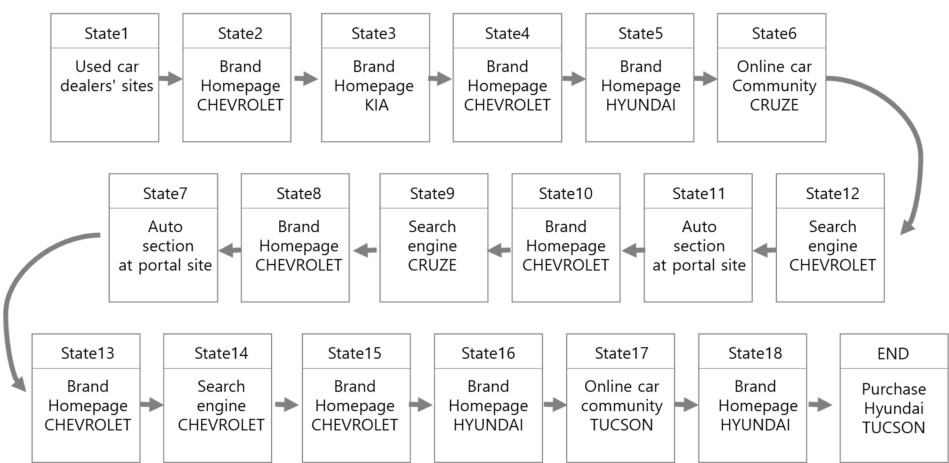

**Figure 2.** An example of a visiting sequence.

### 4.3. Modeling Consumers' Web Search Strategies

The transition matrix is constructed from origin–destination pairs for each movement of each consumer. Analysis of this transition matrix provided a more accurate understanding of the differences in movement between the groups. In this study, as we learned about the CB of each search strategy through the Social Network Analysis (SNA) method using Gephi, we were also able to develop a transition matrix and implement a model for consumer groups' journey flows using R statistics and a visualization program to implement the Markov chains. Figure 3 shows the Markov chain graph of Group 1, which was developed based on origin–destination pair data. Group 1 had a strong tendency to search for information on search engines, automotive sections of portal sites, and brand homepage sites. Group 1 was more likely to visit the homepage of the purchased brand than the homepage of other brands if their journey began with a search engine.

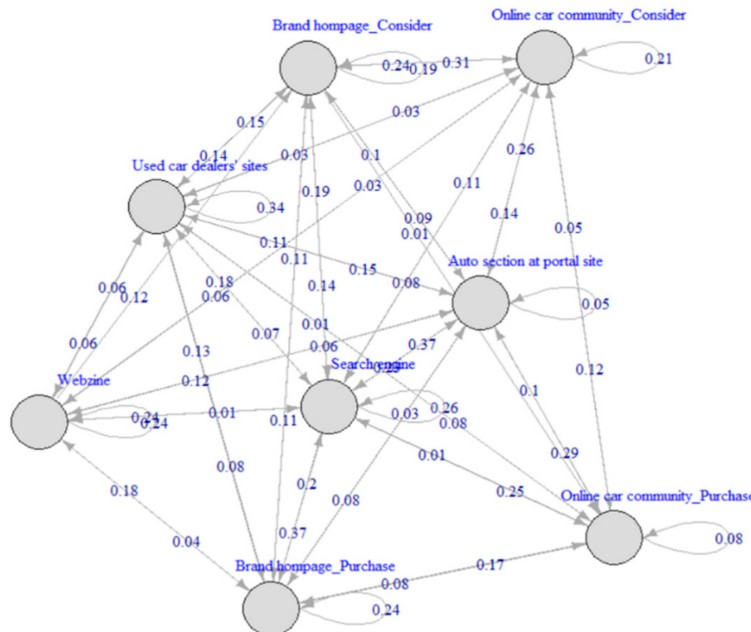

**Figure 3.** Markov chains graph for Group 1.

Figure 4 reveals that Group 2 moved differently to and from search engines and automotive sections of portal sites than Group 1. Group 2 visited online communities more frequently than did Group 1. Group 2 tended to move from search engines to consumer-initiated sites, such as online car communities, rather than to brand-initiated sites. Group 2 tended to search for information about considered brands more than purchased brands, as evidenced by the fact that they moved from the automotive sections of portal sites to the homepage of considered brands or online communities for purchased brands. Group 2 tended to visit sites with abundant information about the brands considered.

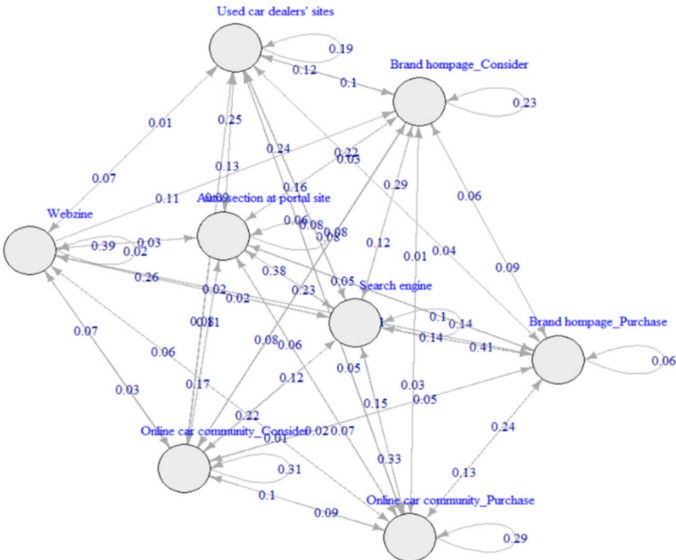

**Figure 4.** Markov chains graph for Group 2.

Figure 5 shows that Group 3 tended to search for information on the automotive sections of portal sites and then move to used car dealers' sites, indicating that they had a third-party-oriented journey. They also frequently moved from search engines to an online community of purchased brands and from search engines, automotive sections of portal sites, and online communities of considered brands to online communities of purchased brands.

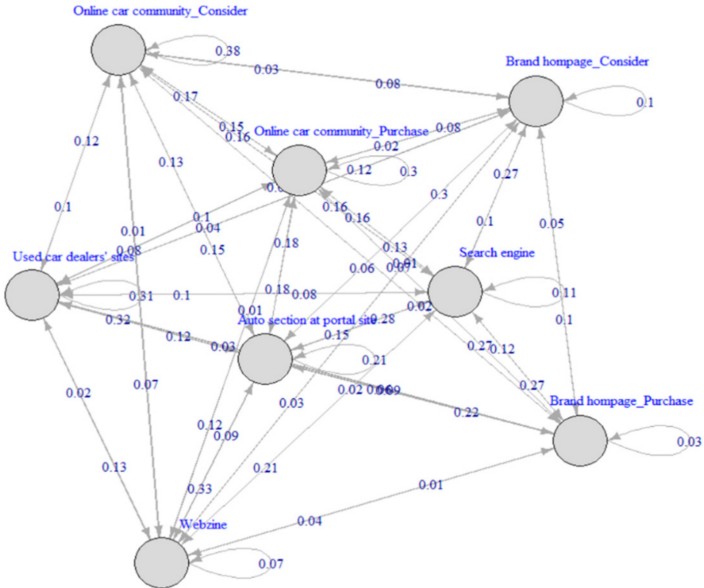

**Figure 5.** Markov chains graph for Group 3.

Groups 2 and 3 had similar consumer movements. However, Group 3 moved more often from search engines to online communities than Group 2 and Group 3 moved from the automotive sections of portal sites to used car dealers' sites more often than Group 2. This difference may be a result of the fact that consumers normally compare products on used car dealers' sites and learn about users' experiences with individual products. Each group exhibited different consumer journey flows. Even if they arrived at the same digital information source, their next destination might differ depending on their group characteristics. This characteristic means that consumer movements affect the length and diversity of consumer journeys. Journey length may be a product of media selection. To clearly determine how the groups selected media differently, the convergence of the groups' transition matrices was analyzed. In addition, we can confirm that the visualization result through the Markov chain graph is very similar to the result of the network analysis using Gephi.

### 4.4. Convergence of Consumers' Web Search Strategy

To understand the search strategy between groups more clearly, we calculated the convergence of the three groups' marginal transition probability using a transition matrix of origin–destination pair sets extracted from visiting sequence data. Table 4 shows the convergence of the probability that the three groups would visit the 13 digital information sources. The sum of the columns in bold numbers is equal to 1. For example, the probabilities that the group with a brand message-oriented search strategy will visit a brand site, a consumer site, or a third-party site were 0.24, 0.20, and 0.56, respectively, the sum of which is 1. With this marginal probability, we conducted an explanatory analysis on how different digital information was used in their path to purchase.

**Table 4.** Marginal probability of each search strategy group.

| Cluster Size | Brand Message-Oriented Search Strategy (n = 23) | Consumer Message-Oriented Search Strategy (n = 27) | Third-Party Message-Oriented Search Strategy (n = 34) |
|---|---|---|---|
| Brand-initiated sites | 0.24 | 0.17 | 0.12 |
| Considered brand homepage | 0.12 | 0.10 | 0.06 |
| Purchased brand homepage | 0.11 | 0.06 | 0.05 |
| Brand microsites or blogs | 0.01 | 0.01 | 0.01 |
| Brand Facebook page | 0.00 | 0.00 | 0.00 |
| Consumer-initiated sites | 0.20 | 0.23 | 0.35 |
| Online community for considered brand | 0.10 | 0.11 | 0.16 |
| Online community for purchased brand | 0.07 | 0.11 | 0.17 |
| Personal blogs | 0.01 | 0.01 | 0.01 |
| General community | 0.02 | 0.01 | 0.01 |
| Third party-initiated sites | 0.56 | 0.60 | 0.53 |
| Search engine | 0.25 | 0.27 | 0.14 |
| Automotive section of portal site | 0.18 | 0.19 | 0.21 |
| Automotive-specific portal site | 0.01 | 0.01 | 0.01 |
| Used car dealer's site | 0.10 | 0.11 | 0.16 |
| Webzine | 0.03 | 0.02 | 0.02 |

Brand-initiated media: Each group had a different marginal probability of visiting brand-initiated media, as shown in Table 4. The marginal probabilities that the group seeking a brand message, the group seeking a consumer message, and the group seeking a third-party message using brand-initiated media were 0.24, 0.17, and 0.12, respectively. The group with the brand message-oriented search strategy was the most likely to use digital information sources in which brand-generated messages were dominant, whereas the group with third-party message-oriented search strategy was the least likely. The group with brand message-oriented search strategy was more than twice as likely to visit

purchased brand homepages than the other groups. This result indicates that the group with a brand message-oriented search strategy moved around the purchased brand's homepages. Furthermore, none of the groups were likely to visit brands' Facebook pages, which are part of the brand's social media presence.

Consumer-initiated media: The marginal probabilities that the group seeking a brand message, the group seeking a consumer message, and the group seeking a third-party message would visit consumer-initiated media were 0.20, 0.23, and 0.35, respectively (Table 4). The group seeking third-party messages was the most likely to visit the online communities of both purchased and considered brands. The probability that the group would visit an online community for considered brands (0.16) was significantly higher than the probability that they would visit a brand homepage of 0.06. This number showed that the group seeking third-party messages tended to seek out consumer-generated messages. Although the group with a consumer-message-oriented search strategy was less likely to visit online communities than the group seeking third-party messages, this group was more likely to frequently visit online communities of purchased brands than purchased brands' homepages. This means that the group with consumer message-oriented groups was less likely to seek a brand message than a consumer message. However, none of the groups were likely to visit personal blogs or automotive bulletin boards of general online communities.

Third party-initiated media: The marginal probabilities that the group seeking a brand message, the group seeking a consumer message, and the group seeking a third-party message would use third-party media were 0.56, 0.60, and 0.53, respectively, as shown in Table 4. The group seeking a brand message and the group seeking a consumer message were more likely to utilize search engines than the group seeking third-party messages, with probabilities of 0.25, 0.27, and 0.14, respectively. The group seeking third-party messages was less likely to use search engines but more likely to visit the automotive sections of portal sites and used car dealers' sites.

### 4.5. Brand Considerations by Search Strategy

RQ2 asked how different the search strategy groups were in the size of the set of brand considerations. Figure 6 shows that the consumer message-oriented search strategy group had a stronger tendency to search for more brands than other strategy groups. There was little difference between the consumer message-oriented search strategy and the third-party message-oriented search strategy, while the number of brands researched by members of the brand message-oriented search strategy group was less than either of the two other groups.

Based on an ANOVA test, the number of brands to be searched was analyzed. Table 5 shows that the differences between the three search strategy groups were significant. This suggests that among the three strategy groups, the consumer message-oriented search strategy group researched the most brands, while the brand message-oriented search strategy group researched the fewest brands.

**Table 5.** ANOVA test of number of brands to be searched.

| Cluster Size | Brand Message-Oriented Search Strategy (n = 23) | Consumer Message-Oriented Search Strategy (n = 27) | Third-Party Message-Oriented Search Strategy (n = 34) | Sig. |
|---|---|---|---|---|
| Number of brands to be searched | 4.2 brands | 7.6 brands | 6.7 brands | ** |

(Significant Level: ** $p < 0.05$).

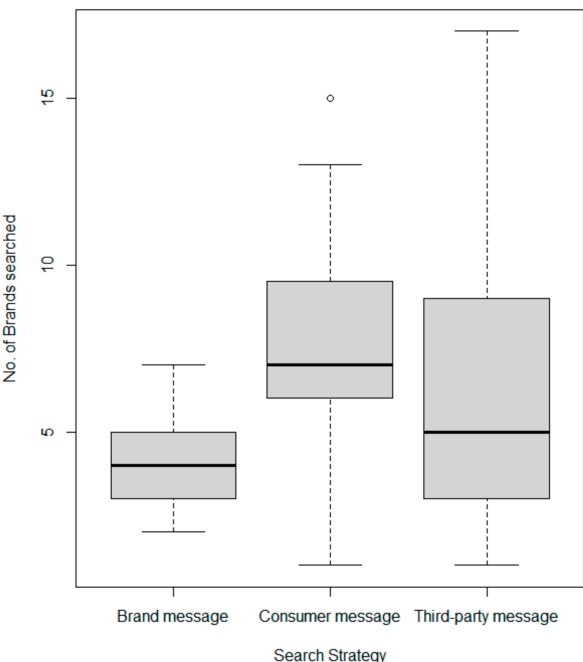

**Figure 6.** Boxplot of number of brands searched by search strategy group.

As shown in Table 6, an ANOVA test showed no difference among the three groups in the number of brands considered during the path to purchase, but the number of brands to be searched was different. This suggests that the consumer message-oriented strategy might lead to learning about more brands than the other strategies. Moreover, considering previous studies, it is plausible that consumer strategy might increase brand uncertainty, especially relative brand uncertainty.

**Table 6.** ANOVA test of number of brands searched versus considered.

| Cluster Size | Brand Message-Oriented Search Strategy (n = 23) | Consumer Message-Oriented Search Strategy (n = 27) | Third-Party Message-Oriented Search Strategy (n = 34) | Sig. |
|---|---|---|---|---|
| From Survey Number of brands considered before starting active research | 2.9 brands | 3.0 brands | 2.6 brands | ns |
| From PC Clickstream Number of brands to be searched | 4.2 brands | 7.6 brands | 6.7 brands | ** |

(Significant Level: ** $p < 0.05$)

## 5. Conclusions

We aimed to understand how different web search strategies influenced the size of brand considerations, and the results indicated five main findings. The first is that consumers searched for information around the core web platforms; thus, three web search strategies, namely, brand messages, consumer messages, and third-party messages, were identified differently by the main information source. Second, there was a slight difference in gender distribution among these three groups in that more male buyers were included in the consumer message-oriented search strategy group. Moreover, there were significant differences in the automobiles they bought in terms of brand origin. The third finding is that even though the process increased as the number of brands investigated during the search

process increased, not all potential buyers changed their behavior to fit the same pattern. We proved that, according to the three search strategies, the number of times consumers consider brands can differ. That is, we concluded that consumer-message-oriented or third-party message-oriented search strategies can increase the number of brands in the set of considerations. Fourth, we found that different groups of consumers directly surveyed the brand launch site to learn about the target brand, reducing uncertainty about individual brands. However, some consumers undertook a random search and had longer journeys. We noticed search strategies primarily as efforts to reduce perceived risk. The third-party platform provided an efficient comparison instrument, which suggested a new brand as a candidate and results in an iterative web search for reducing uncertainty and risk. Likewise, the selection of a digital information source can transform the evaluation stage of the purchase process. Finally, we found practical implications for marketing practitioners' strategies and consumer choices through this study. Consumers' digital media choices during web searches affected the number of brands they consider, which in turn affected the length and complexity of their journey to a decision.

This study also has practical implications for companies and governance-related public institutions. First, we suggest that brand websites are important for retaining loyal customers who do not have brand uncertainty, whereas consumer-initiated or third-party-initiated websites play an important role in attracting competitors' customers. Second, companies or related government agencies can use the consumer message-oriented or third-party message-oriented search strategies presented in this study to lead online or mobile consumer searches in a more diverse and rational way. These strategies allow consumers to search for a variety of brands and increase the overall web duration time to increase the probability of consumer purchase. Third, direct research about brands on various websites with many consumers and searching for the target brand have a significant effect of reducing uncertainty at the individual level, and likely have an advantageous impact for enterprises as well. Companies can try to find ways to expose more of their products to online or mobile sites through a variety of channels, which can positively affect the overall image of their own brand and increase consumer preference. Finally, we found that consumers' digital media choices influenced their journey to make brand decisions. Therefore, both companies and government agencies in charge of related governance (e.g., legal sanctions on online websites) should make more efforts to construct a simple UI and a complex system for consumers from which to choose. We believe that when such an underlying platform is formed, channels that can satisfy the diverse needs of consumers can be completed.

Despite the many suggestions of this study, it also has limitations. First, in this study, clickstream data did not include information search data from mobile devices. Future research should consider search behavior on mobile platforms, which will yield more realistic conclusions. The second limitation was that this study focused solely on consumers' decisional processes in automobile purchases, which are highly involved. Thus, this study's findings cannot be applied to low-involvement purchases, which may have shorter journeys and require less searching for information. Future research should attempt to generate more generalizable findings by examining search behaviors for a diverse range of products and product categories to formulate the more robust categorization and mapping of consumers' decisional processes. Third, this study did not account for searches for offline information, such as information obtained from automobile dealers, friends, and relatives. Therefore, future studies should consider offline information search processes to generate more realistic findings and applicable guidance. Fourth, this study was conducted based on data from online websites. However, we were not able to separate IR data from mobile application data, which has become more commonly used by customers in recent years. Therefore, we must consider the study of mobile application data in future research. Finally, there is a lack of reference to research on how consumer ethnocentrism influences the choices made. As there have been many preceding studies on differences based on

national orientation, we hope that research on national and ethnic-biased search tendencies will be conducted in the future.

**Author Contributions:** Data curation, S.K., Z.L., and J.K.; formal analysis, S.K., Z.L., and J.K.; methodology, S.K., Z.L., and J.K.; project administration, Z.L.; software, S.K., and J.K.; validation, J.K.; writing—original draft, S.K., Z.L., and J.K.; writing—review and editing, J.K. All authors have read and agreed to the published version of the manuscript.

**Funding:** This research received no external funding.

**Institutional Review Board Statement:** Not applicable.

**Informed Consent Statement:** Not applicable.

**Conflicts of Interest:** The authors declare no conflict of interest.

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
