# Peer review of "Advances in Search Strategy Using the Set of Brand Considerations in the Web Ecosystem"

_applsci, doi:10.3390/app11083514_

Round 1

Reviewer 1 Report

First of all, I must say the paper is very well written, coherent, having a very logical flow, and being easy to read and understand. Also, I must express my appreciation for the work load assumed by the authors, who gathered, structured and analyzed a significant amount of data, using various, but appropriate data analysis techniques.

However, there are two main drawbacks of the paper which I believe should be addressed in order for the paper to be publishable.

Firstly, despite the huge amount of preliminary data which was involved in the research, the actual study sample is extremely small: 84 consumers. Further on, various more or less complex methods are employed to analyze clusters of consumers, which are even smaller: 23, 27, and respectively 34 consumers. With all due respect, any results yielded by such small samples are pretty much insignificant, the study being rather an exercise that demonstrates how such an analysis should be conducted. Unfortunately, despite the quality of the paper in what concerns the narrative, in order for the study to be scientifically valid and publishable, the sample should be significantly larger. As I see it, there are two options for the authors:

- Option 1: Investing more research resources to gather more clickstream data and repeat the whole data structuring and analysis process.

- Option 2: Coming up two categories of arguments: on one hand, authors need to emphasize methodological aspects that validates results based on such small samples, for the employed methods; on the other hand, authors should point out previous research using similar methods for similarly small samples.

Secondly, although the paper’s main focus is to provide practical implications for marketers, the actual practical implications described in the paper are confined to a single paragraph: “We suggest that brand websites are important to keep loyal customers who do not doubt relative brand uncertainty, whereas consumer-initiated or third-party-initiated websites play an important role in attracting competitors’ customers”. Of course, one might argue that the actual results bring practical implications by themselves, but that must be clearly stated: marketers should …, companies should etc. What I suggest is that authors create a distinct section entitled “Practical implications” where these issues be described extensively.

Lastly, there is a minor issue with the acronyms: IR and CB are used, but not defined anywhere in the text.

Reviewer 2 Report

Please find attached the Review Report.

Reviewer 3 Report

Please analyse the text in terms of style, punctuation and eliminate typos. Besides, consider that if the author writes the text in British English, one should follow that style and not mix it with American English. 

Please consider that apart from the practical part, scientific research should also develop theories. The article would gain cognitive value if the author emphasized more which part of the theory the author intends to develop thanks to this research in the introduction.

One may emphasize that the author pointed to the limitation of the research. A significant limitation is that clickstream data did not include information search data from mobile devices. Meanwhile, this seems to be the primary source of search data. There is also a lack of reference to research on how consumer ethnocentrism influences the choices made. Many articles on this subject also include studies on public ethnocentrism, cognitive orientation, and preventive measures. Perhaps it is worth pointing to this as an area for further research? Regardless of these comments, one can congratulate the author on exciting research. However, please indicate more precisely how this research contributes to the development of the theory and to what extend. 

Round 2

Reviewer 1 Report

The authors reasonably addressed both my major concerns about the paper: the first one, regarding the investigated sample size, and the second one, concerning the practical implications of the study. From my point of view, the revised version of the paper meets the standards needed for it to be considered publishable.

Author Response

We would like to thank you for your comments. We are very pleased that your suggestion seems to have further advanced our research. In addition, as you suggested, we again received the English proofreading service to make the English expressions of our study more readable and clear meaning commentary for many readers.

Reviewer 2 Report

The revised version of the submission titled “Advances in Search Strategy using the Set of Brand Considerations in the Web Ecosystem” improved in a positive manner. The author(s) replied to all the suggestions and recommendations formulated during the previous review round. Therefore, the quality of the paper enhanced significantly. Henceforward, I recommend paper acceptance in current form.

Author Response

(The authors gave the same response as above.)
